# Food Safety Challenges and Barriers in Southern United States Farmers Markets

**DOI:** 10.3390/foods9010012

**Published:** 2019-12-21

**Authors:** Zahra H. Mohammad, Heyao Yu, Jack A. Neal, Kristen E. Gibson, Sujata A. Sirsat

**Affiliations:** 1Conrad N. Hilton College of Hotel and Restaurant Management, University of Houston, Houston, TX 77204-3028, USA; zhmohamm@Central.UH.EDU (Z.H.M.); JNeal@Central.UH.EDU (J.A.N.); 2School of Hospitality Management, Pennsylvania State University, 230 Mateer Building University Park, PA 16802, USA; chandler.yuheyao@gmail.com; 3Department of Food Science, Center for Food Safety, Division of Agriculture, University of Arkansas, Fayetteville, AR 72704, USA; keg005@uark.edu

**Keywords:** farmers market, food safety, barriers, farmers market managers, farmers market vendors

## Abstract

Purchasing fresh and local produce at farmers markets has seen an increasing trend over the past decade. However, with this rise in popularity food safety challenges need to be recognized and addressed. Farmers market managers play a significant role in ensuring that vendors implement food safety practices at the market. Thus, this study investigated the food safety perceptions of farmers markets managers and vendors in Texas and Arkansas. A total of 123 participants were surveyed, including 38 managers and 85 vendors. The survey included a series of questions to determine gaps in vendor and manager food safety knowledge as well as the barriers and factors that prevent the implementation of relevant food safety practices. The results indicate that a lack of facilities, equipment, and resources containing food safety guidelines specific to farmers markets were the major barriers to the implementation of food safety practices. In addition, only 36.7% of participants had formal food safety training (e.g., ServSafe) and approximately 50% of the market managers provided food safety materials to their vendors. Overall, these data suggest that the development of farmers market-specific training programs to enhance food safety behaviors and practices in farmers markets would be beneficial to stakeholders.

## 1. Introduction

The number of farmers markets have increased by 300% in the United States (U.S.) from 2863 markets reported in 2000 to 8720 farmers markets in 2017 [1]. In 2014, Texas and Arkansas were among the top ten states with the largest increases in the number of farmers markets [2], and by 2018, Texas and Arkansas had 226 and 111 farmers markets, respectively [1]. Ready-to-eat food products, primarily fresh fruit and vegetables, usually dominate the available goods at these markets across the country [3,4,5,6]. With the growing number of farmers markets, consumers’ demand for locally grown fresh fruit and vegetables has also increased [7]. This rise in fresh produce consumption may be attributed to an increased awareness about nutrition and enhancing healthy eating habits [8]. Studies have shown that consumers have positive assumptions about the quality and safety of food sold at farmers markets [9,10,11]. A previous survey-based study reported that approximately 60% of the respondents agreed that locally grown produce are safer than conventional produce [10]. From a socioeconomic perspective, farmer’s markets enhance economic growth in the United States through selling locally grown foods and increasing job opportunities. In addition, since both governmental policies and media support local food, farmers’ markets promote economic development for rural and urban communities [12]. However, microbial studies have found that the quality and safety of food products sold at farmers markets are not always better than food at retail stores, and in some cases, the microbial levels in produce from farmers markets were higher than retail establishments [13,14,15,16]. According to the U.S. Centers for Disease Control and Prevention (CDC), foodborne illness outbreaks cause an estimated 48 million illnesses, 128,000 hospitalizations, and 3000 deaths in the United States annually [17,18,19]. Fresh fruits and vegetables-responsible for 46% of the foodborne illnesses [20,21]—cause the largest number of illnesses compared to any other food category, as these foods typically receive no additional kill step such as heat treatment during processing [6,22,23,24,25]. While there are few outbreaks associated with foods sold at farmers markets, it is important to recognize that a majority of foodborne illness outbreaks go unreported [6]. Moreover, there is limited traceability on produce and other food items sold at farmers markets [26].

Products sold at farmers markets have been associated with at least ten foodborne illness outbreaks in the U.S. [27]. For instance, an outbreak of *Campylobacter jejuni* was linked to the consumption of raw peas at a farmers market in south-central Alaska in 2011 [28]. An *E. coli* O157:H7 outbreak was linked to the consumption of strawberries sold at the farmers market and roadside stands in Oregon in 2011 [29], and one case of salmonellosis was linked to consumption of tomatoes purchased from a farmers market [30]. In addition, multiple outbreaks of foodborne illness have been associated with non-produce products sold at farmers markets in the United States and Canada [31,32,33]. Of these, outbreaks of salmonellosis, *E. coli* O157:H7 hemorrhagic colitis, and listeriosis were implicated with different kinds of cheese sold at farmers markets [32,33].

Major food safety risk factors at farmers markets include improper handling, temperature abuse, contaminated equipment, and poor personal hygiene [34,35]. Some of the risks can be linked to the nature of operations at farmers markets. More specifically, markets are typically located outdoors in temporary locations which lack the necessary equipment and facilities (e.g., refrigeration and handwashing stations) to implement basic risk management practices and are usually exempt from state or local food safety regulations [6,34,36]. Therefore, it is critical to equip farmers market managers and employees with science-based strategies to enhance food safety. In addition, farmers markets managers’ knowledge and beliefs about food safety are essential for proper handling and implementation of food safety practices by the vendors [37,38,39].

Training and educational programs along with good manufacturing practices (GMP) and good handling practices (GHP) are critical to reduce the risk of foodborne illness [40]. Manager behavior and willingness to offer appropriate resources and food safety training to the vendors is important [37]. In the case of farmers markets, the manager may not have formal food safety training, and nearly all farmers market vendors are exempt from compliance with the Produce Safety Rule within the Food Safety Modernization Act (FSMA) based on the $25,000 annual produce sales cut-off [3,41]. Food safety related perceptions, attitudes, and practices of farmers markets managers, vendors, and customers have been documented across the U.S. and other countries [11,34,36,38,42,43,44,45,46,47,48,49,50]. Behnke, Seo, and Miller [34] assessed the food safety practices at farmers market and reported that managers and vendors did exhibit knowledge of food safety. However, the results revealed that some vendors were not able to implement food safety practices because of a lack of resources such as handwashing stations [34].

To date, there is no published research on the knowledge and perceptions of food safety at farmers markets in Texas and Arkansas nor on the barriers to implementation of best practices at farmers markets in these states. Thus, the primary objectives of this study are to (1) investigate the food safety perceptions of farmers markets managers and vendors in Texas and Arkansas; (2) characterize current food safety content accessible to them; (3) determine the challenges and barriers to implementation of food safety practices; and (4) identify the critical gaps that need to be addressed.

## 2. Materials and Methods

### 2.1. Participants and Data Collection

A survey instrument was designed to characterize farmers market manager and vendor food safety perceptions. Questions were screened and approved by the University of Houston’s Human Subjects Review Board as well as the University of Arkansas’s Institutional Review Board. The surveys were disseminated at farmers market meetings in TX and AR. The participants included 38 managers and 85 vendors. The survey was disseminated by investigators at farmers market conferences across TX and AR. The sample size was determined based on the margin of error from the confidence interval based on population distributions in the areas where the survey was conducted.

### 2.2. Measures

Farmers markets managers and vendors received different versions of the questionnaire. Both manager and vendor questionnaires included four sections to obtain similar information. The questions in the Section 1 of both surveys assessed participants’ food safety knowledge and self-reported food safety practices. Both constructs were measured by a five-point Likert scale (1 = strongly disagree, 5 = strongly agree). General food safety knowledge was measured using four items adopted from a previous study by Yu et al. [11]. An example question was “I think unsafe food can make people really sick.” Food safety practices were measured using four items also adopted from Yu et al. [11]. An example of food safety practice items was “I can clean and sanitize equipment and utensils properly.”

The questions included in the Section 2 of both surveys were designed to examine the barriers for market vendors and managers to implement proper food safety handling procedures at farmers markets. These barriers were adopted from a previous study by Yapp and Fairman [51] and included: (1) lack of infrastructure; (2) lack of benefits to the vendors and managers to implementing food safety practices; (3) high costs; and (4) lack of specific food safety guidelines for farmers markets to follow. All of the items were measured using a five-point Likert scale (1 = strongly disagree, 5 = strongly agree). An example question was “I feel that practicing safe food handling procedures at markets is too costly.”

The Section 3 of the vendor survey contained questions about current food safety training at farmers markets to obtain the information on the following: (1) whether managers provide food safety training material; (2) food safety policies (e.g., glove usage, providing samples, etc.) at their markets; and (3) vendors’ perception of current food safety materials. All the items were measured using a five-point Likert scale (1 = strongly disagree, 5 = strongly agree). The Section 3 of the manager survey included questions related to (1) whether managers provided food safety materials to their vendors; (2) content included in current food safety training and policies; (3) managers’ perception of current food safety materials; and (4) whether managers have a designated food safety manager at their markets.

The final section on both surveys included questions to obtain demographic information of participants. Participants were requested to report general socio-demographic questions including gender, age, and education level and specific questions related to their work position such as their job titles, tenure, and the number of markets they attended as vendors/managers.

### 2.3. Data Analysis

Separate descriptive analyses were used to acknowledge current food safety training programs at farmers markets and the barriers to implementing proper food safety handling procedures using JMP (Version 14, SAS Institute Inc., Cary, NC, USA). The descriptive analysis was used to calculate mean and standard deviation for each variable including manager and vendors responses to food safety training programs and the mean of responses versus barriers to implementing food safety practices. Analysis of variance (ANOVA) was conducted, and a 95% confidence level (α < 0.05) was used for statistical significance to compare whether vendors and managers perceived these barriers differently. In addition, hierarchical multiple regression was conducted to examine factors influencing participants’ self-reported food safety practices. IBM SPSS Statistics 22.0 for Windows (IBM Corporation, Armonk, NY, USA) was used to analyze the data.

## 3. Results and Discussion

### 3.1. Demographic Analysis

Table 1 outlines the socio-demographic factors of the survey respondents (N = 123). The respondents included 38 farmers markets managers (18% male and 82% female) and 85 vendors (38% male and 62% female). In terms of generation, the majority of the participants were baby boomers (64%), while only 21.1% and 26.8% of the participants were millennials and generation X, respectively. More than 35% of the respondents had a bachelor’s degree, and approximately 30% of the participants had graduate or post-graduate degrees. In terms of the products that participating vendors sell at farmers’ markets, 82% of vendors sold fresh produce while 6% of the vendors sold fresh meat. Therefore, produce was represented as the majority of products sold at surveyed farmers markets. Majority of farmers markets had a U-shaped layout.

### 3.2. Current Food Safety Training and Policies at Farmers Markets

Questions related to individual food safety training and policies at farmers markets were included in the survey. Data related to food safety training and policies at farmers markets are presented in Table 2. Only 37.6% of market vendors reported that the managers require glove usage while handling ready-to-eat foods, and 43.5% of the vendors reported that the managers required them to provide utensils or toothpicks for food samples. The Food Code is a model for each state to adopt; however, states are not required to follow these recommendations. According to the FDA 2017 Food Code, a food handler should never handle ready-to-eat foods with bare hands and should wash hands thoroughly before putting on gloves and when changing to a new pair [52]. Meanwhile, 50.0% of the market managers reported that they required their vendors to use disposable gloves and provide utensils for food items, respectively.

In terms of food safety training at farmers markets, the data indicate that only 45 respondents (36.7%) including 15 managers and 30 vendors had received formal food safety training (e.g., Servsafe^®^ Food handler course, general food safety training including handwashing procedure and time/ temperature abuse, and GAP training), previously. When asked whether food safety guidelines or training were provided for their farmers markets vendors, 50% of the markets’ managers indicated that they provide guidelines or training to their vendors, and 50.6% of the vendors reported that they had received food safety training from their market managers in the past. The consistent results from vendor and manager surveys indicated that approximately half of the farmers markets managers provided some type of food safety training material.

Among the managers who provided guidelines or outreach information to their vendors, 17 managers (89.4 %) provided handwashing training, and 8 managers (42.1%) provided training on standard operating procedures (SOPs) for facilities and equipment such as handwashing stations and refrigeration or cooling units. Among those vendors who reported that managers provided training materials, 37 (82.2%) indicated that managers provided training materials about handwashing, while only 12 (28.0%) vendors indicated that the training materials related to SOPs of facilities and equipment were provided by their managers.

In terms of the evaluation of current food safety training material at farmers markets, only 18 (42.0%) out of the 43 vendors who received food safety training material from managers before indicated that they had received high-quality training materials and the contents of the training material were comprehensive (Table 2). In addition, among the managers who reported that they provided food safety guidelines for their vendors, only 9 (47.4%) out of 19 managers believed that they provided high-quality materials, and 7 (36.8%) reported that the content of the training material was comprehensive.

According to these results, the majority of market managers and vendors never received any formal food safety training, and approximately 50% of the market managers provide food safety guidelines or training materials to their vendors. A majority of the managers surveyed do not require vendors to use disposable gloves and do not provide SOP for facilities and equipment. These findings are similar to those reported by Harrison et al. [44], who surveyed food safety practices on small and medium-sized farms and farmers markets in Georgia, Virginia, and South Carolina. The researchers reported that more than 42% of managers did not have specific food safety standards for their farmers markets. Both Harrison et al. [44] and the current study highlight gaps in farmers markets food safety standards and guidelines that may lead to an increase in potential contamination of food products sold at these markets and thus pose a risk to consumers.

### 3.3. Food Safety Knowledge of Farmers Market Managers and Vendors

In terms of food safety knowledge of farmers markets managers and vendors, the means of 83 respondents included 38 managers and 45 vendors (Table 3) indicate that 92% managers (mean = 4.58), and 89% vendors (mean = 4.44) agreed that foodborne illness could be life-threatening. Approximately half of the managers and vendors agreed that the probability of developing a foodborne illness is minimal, and some think that food is always safe to eat. Additionally, 91.6% of farmers markets managers (mean = 4.47) and 89% of vendors (mean = 4.44) believe that proper food safety practices reduce both food poisoning and bacterial growth. In addition, when farmers markets manager and vendors asked about how they feel confident about what they are doing at farmers markets, 42% of managers (mean = 4.29) and 44% of vendors (mean = 4.33) reported that they feel confident in their ability to store or transport food at a safe temperature. Furthermore, 55% of managers (mean = 4.38) and 49% of vendors (mean = 4.4) reported that they feel confident in their ability to clean and sanitize equipment and utensils properly. Finally, according to the data, 70% of managers (mean = 4.63) and 60% of vendors (mean = 4.56) exhibited confidence in their ability to learn proper food safety practices. Although there was some variation in responses from farmer markets managers and vendors, no significant differences were found between managers and vendors responses in terms of their knowledge of food safety.

Based on the results of the current study, farmers markets managers and vendors possess general food safety knowledge. However, there has been evidence of poor safety practices in farmers markets. Norwood et al. [53] conducted food safety observations among 300 vendors in farmers markets and reported multiple high-risk practices such as pets defecating around booths, vendors reusing cardboard produce boxes, live farm animals (goats) present, and no access to restroom facilities for vendors and patrons [53]. Behnke, Seo, and Miller [34] assessed food safety behavior at farmers markets and reported that although managers and vendors demonstrated some food safety knowledge, there was a gap in implementing food safety practices. According to Worsfold, Worsfold, and Griffith [41], who surveyed farmers markets vendors in the U.K., about 25% of vendors lacked food safety knowledge, and almost 84% of them believed that their products are safe and do not cause foodborne illness. Hence, farmers market vendors and managers may benefit from standardized formal training in order to increase their awareness about food safety.

These studies highlight the need for training and awareness about the importance of food safety practices.

### 3.4. Barriers to Implementing Proper Food Safety Handling Practices

Managers and vendors were specifically asked about the role of four barriers in their ability to implement safe food handling practices at farmers markets including (1) lack of proper facilities and equipment; (2) lack of specific food safety guidelines for farmers markets; (3) cost of implementation; and (4) lack of benefit to their business, for example increasing the amount of sales or attracting more customers.

A two-way ANOVA with four barriers and job positions (i.e., vendor and manager) as independent variables was conducted to compare participants’ perceptions about the barriers with the implementation of food safety handling procedures at farmers markets. The results of the two-way ANOVA indicate that there were significant differences among participants’ perceptions of four barriers (*F* (3, 383) = 7.21, *p* < 0.01) (Table 4). However, there was no significant difference between managers and vendors in terms of the perception of the four barriers (*F* (1, 385) = 1.64, *p* > 0.05). In addition, the interaction effect of barriers and job position was also insignificant (*F* (3, 383) = 1.10, *p* > 0.05). According to the Tamhane post hoc test for the four categories, the majority of participants agreed that the lack of proper facilities and equipment is the greatest barrier, while lack of specific food safety guidelines for farmers markets was the second most significant barrier. The participants believed that both the cost and lack of benefit to the market or their business was a less significant obstacle to the implementation of food safety practices (Figure 1). The independent sample *t*-test indicated that there were no significant differences (*p* > 0.05) between market managers and vendors in terms of their perceptions about these barriers.

Farmers market vendors may not undergo formal inspection unless by choice or required by a buyer, though they could have GAP training and certification. Based on this, there are potential gaps in food safety practices along with a possible increased risk for contamination within farmer markets [26,34]. Most farmers markets do not have proper handwashing facilities, refrigeration, and restrooms because these venues are usually sited in transient locations or in a municipality’s green area where it can be a challenge to provide these resources [37]. Regardless, these resources should be a priority for both farmers market managers and market boards, if applicable, in order to support the implementation of effective food safety practices. A previous study has investigated the relationship between the growing number of farmers markets and the number of cases of foodborne illness in the United States and found a positive correlation between this increase and norovirus-related outbreaks [54]. This may be due to the lack of handwashing stations and refrigerators accessible at farmers markets. Behnke, Seo, and Miller [34] examined farmers markets in Indiana and indicated that only 9 out of 18 (50%) vendors had access to functional handwashing stations. Recent studies conducted in North Carolina and Virginia farmers markets reported that about 27% and 40%, respectively, of observed markets did not have handwashing facilities and about 87% of those that had handwashing facilities did not have illustrations for proper handwashing [47,55]. The CDC [56] has recognized that poor personal hygiene and sanitation by food handlers is one of the most critical risk factors that increase the likelihood of foodborne illness.

Lack of specific food safety guidelines for farmers markets to follow was found to be the second major barrier to implementing food safety practices at farmers markets. The food safety guidelines and best practices available for food handlers and food safety purposes are usually general, which include information and standards that are not applicable to farmers markets vendors and managers. Therefore, it may be beneficial for state cooperative extension and regulatory bodies to work together to facilitate uniformed and effective facilities and equipment as well as high-quality training designed specifically for farmers markets managers and vendors.

### 3.5. Factors Influencing Self-Reported Food Safety Handling Practices

Factors influencing market managers and vendors’ food safety handling practices were examined by hierarchical linear regression using ordinal least square (OLS). In the regression model, the dependent variable was self-reported food safety handling practices rated on a five-point Likert scale. Two demographic factors, age and gender were included in the first regression model as control variables. In addition, participants’ job position (i.e., manager and vendor), food safety training (coded as 1 = received training, 2 = never received training), and participants’ food safety perceptions were included in the second regression model after controlling for age and gender. The results are shown in Table 5. The first regression model was not significant overall (*F* (1, 121) = 0.80, *p* > 0.05). Both gender (*B* = −0.03, *p* > 0.05) and age (*B* = 0.01, *p* > 0.05) were not significantly related to respondents’ reported food safety practices. The second model included three independent variables (i.e., job position, food safety knowledge, and food safety training) and the model was significant overall (*F* (5, 117) = 5.57, *p* < 0.01) and the determination coefficient (R^2^) was 0.28, indicating acceptable model fit. The results of model 2 revealed that participants’ food safety perception was significantly related to their reported food safety practices (*B* = 0.34, *p* < 0.01). This indicates that the participants with high food safety perception tended to report performing proper food handling practices. Additionally, managers’ food safety practices were significantly higher than vendors’ level (*B* = 0.48, *p* < 0.05). However, whether or not participants completed formal food safety training did not influence their reported food safety practices (*B* = 0.10, *p* > 0.05). This may indicate that general food safety training programs (e.g., ServSafe) may be ineffective for farmers market vendors and managers. These results are in agreement with those reported by Worsfold, Worsfold, and Griffith, [41] who conducted a study at a single farmers markets in the United Kingdom and found most of the vendors had food safety training previously, but less than 50% of these vendors did not consider their products could be contaminated. Although studies have demonstrated that training may increase food safety knowledge, this does not always translate to positive food safety behavior and attitudes unless the perception is changed [40,57]. Enhancing food safety behaviors should not be limited to farmers market managers and vendors alone. Consumer’s behavior also plays a significant role in increasing the risk of foodborne illness. Studies have shown that enhanced consumer food safety practices may lead to a decrease in the risk of foodborne illness at farmers markets [10]. Wolf, Spittler, and Ahem [58] investigated consumers’ perception and the reasons they shop at farmers markets. The authors found that quality and value held the most important attributes for consumers to purchase at farmers markets. Consumers perceive that the produce sold at farmers markets look fresher, are higher-quality, and a better value or reasonably priced, but they do not necessarily consider safety of the products. Multiple studies have found that there is a strong relationship between food safety perception and practices [44,49,59,60]. To this end, changing the perception and attitude of farmers markets managers and vendors through specific and high-quality training should be a priority for reducing the potential risk of foodborne illness from these markets [61]. Choi and Almanza [62] suggested that training for farmers markets operators should be different from that provided for foodservice employees and focused on the products sold at these markets as well as personal hygiene, proper handling, and food safety practices. Ultimately, food safety at farmers markets is achieved by translating behavior into food safety culture [63]. This culture should include farmers market managers, vendors, and consumers.

## 4. Conclusions

The results of this study suggest that there are deficiencies in information and practices as reported by both managers and vendors. Overall, the data indicated that farmers markets managers and vendors are in need of improvement in terms of training or outreach programs to increase their perception of food safety risks at the market and facilitate the implementation of best practices. The present study also found that barriers perceived by farmers market managers and vendors to be obstacles to the implementation of food safety practices include a lack of facilities and equipment and lack of specific food safety guidelines for farmers markets to follow. These reported barriers should be addressed to encourage the implementation of food safety practices at farmers markets in Texas and Arkansas. Our findings support that there is a need for further food safety training and educational outreach tailored for farmers markets to enhance the safety of foods sold and to positively impact food safety behaviors of farmers markets vendors and managers.

## Figures and Tables

**Figure 1 foods-09-00012-f001:**
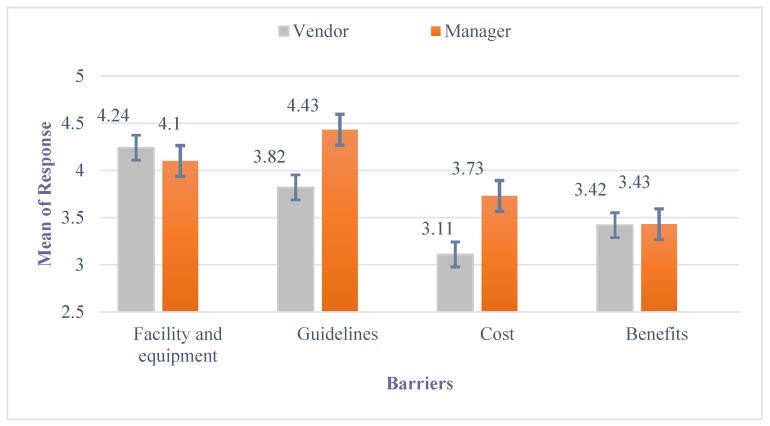
Barriers to implementation of safe food handling practices at farmers markets. Strongly disagree = 1, strongly agree = 5, the error bar denoted the range of 95% confidential interval. Error bars represent the ± standard deviation of average of responses of manager or vendor to each question.

**Table 1 foods-09-00012-t001:** Demographic profile of the respondents (N = 123).

Characteristics	N (Frequency %)
**Gender**	
Male	40 (32.0)
Female	83 (68.0)
**Generation**	
Baby boomers (52 years above)	64 (52.1)
Generation X (37 years to 51 years)	33 (26.8)
Millennial (18 years to 36 years)	26 (21.1)
**Education Level**	
High school degree	28 (22.1)
Associate degree	16 (13.1)
Bachelor’s degree	43 (35.2)
Graduate degree	30 (24.6)
Post-graduate degree	6 (4.9)
**Food Safety Training**	
Yes	45 (36.9)
No	78 (63.4)
**Position**	
Manager	38 (4.9)
Vendor	85 (62.9)
**Vendors’ Primary Product**	
Fresh meat	5 (5.9)
Produce	70 (82.4)
Ready-to-eat food	2 (2.4)
Other	8 (9.5)

**Table 2 foods-09-00012-t002:** Current food safety training and policies at farmers markets.

	Manager (Frequency %)	Vendor (Frequency %)
Food safety course	15 (39.5)	30 (35.3)
Disposable gloves	19 (50.0)	32 (37.6)
Utensils for food sample	19 (50.0)	37 (43.5)
Food safety materials	19 (50.0)	43 (50.6)
*^a^* SOPs	8 (42.1%)	12 (28.0)
Handwashing	17 (89.4%)	37 (82.22)
*^b^* Good quality	9 (47.4%)	18 (42.0%)
*^c^* Comprehensive	7 (36.8%)	18 (42.0%)

*^a^* This refers to a question of whether the manager provided guidelines such as standard operating procedures (SOPs) for facilities and equipment such as handwashing stations and refrigeration or cooling units. *^b^* Refers to a question of whether the manager provided high-quality food safety outreach training. *^c^* Refers to a question of whether the manager provided the training material with comprehensive food safety content.

**Table 3 foods-09-00012-t003:** Mean of responses of self-assessment of food safety knowledge at farmers markets.

Statements	Manager *^a^* (Response)	SD *^b^*	Vendor *^a^* (Response)	SD *^b^*
Foodborne illness can be life threatening	4.58 *^c^*	0.8	4.44	1.0
A foodborne illness could be dangerous	4.71	0.6	4.49	1.0
I think unsafe food can make people really sick	4.57	0.8	4.40	1.0
The odds of developing a foodborne illness are very small	2.63	1.1	2.47	1.1
I think food is always safe to eat	1.87	0.8	1.71	0.9
Proper food safety practices reduce food poisoning	4.48	0.6	4.36	0.8
Proper food safety practices reduce bacteria growth	4.46	0.7	4.44	0.6
I feel confident in my ability to store/transport food at safe temperatures	4.29	0.7	4.33	0.7
I feel confident in my ability to clean and sanitize equipment and utensils properly	4.38	0.8	4.40	0.7
I feel confident I can learn proper food safety practices	4.63	0.6	4.56	0.6

*^a^* Response refers to the average responses of agreement with respect to each question. *^b^* SD refers to a standard deviation for manager or vendors’ responses. *^c^* No significant differences (*p* > 0.05) were found between manager and vendors responses to all of food safety knowledge questions.

**Table 4 foods-09-00012-t004:** Results of two-way ANOVA on identified four barriers of implementing proper food handling practices.

Source	Sum of Squares	*df*	F-Value	*p*-Value
Barriers	21.89	3	7.21	0.00
Job position	1.67	1	1.64	0.20
*^a^* Barriers position	3.35	3	1.10	0.35

*^a^* this indicates that there were no significant differences (*p* > 0.05) in perceptions of each of identified barrier between managers and vendors.

**Table 5 foods-09-00012-t005:** The hierarchical linear regression on food safety practices.

Predictor	Food Safety Practices	
Model 1	Coefficient	*S.E.*
Gender	−0.03	0.22
Age	0.01	0.01
*R* ^2^	0.05	
Model 2		
Gender	−0.11	0.21
Age	0.01	0.01
Job position	0.48 *	0.21
Food safety knowledge	0.34 **	0.11
Food safety training	0.10	0.14
*R* ^2^	0.28 **	
Δ*R*^2^	0.23 **	

* *p* < 0.05; ** *p* < 0.01.

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
