# Peer review of "Food Safety Challenges and Barriers in Southern United States Farmers Markets"

_foods, 2019, doi:10.3390/foods9010012_

Round 1

Reviewer 1 Report

The article presents an interesting issue of food security in the context of selling food at a local farmers market in Texas and Arkansas. In particular, the Authors focused on assessing attitudes and perceptions of farmers and market managers and vendors and characterized the basic challenges and barriers in the implementation of food safety practices. The research is important in the light of the presented diagnosis of threats resulting from the lack of appropriate procedures and behaviors of market participants. Study results can improve the system of monitoring and prevention not only in the analysed regions, but also outside of the states borders. One of the key challenges, as the Authors indicate, is to create an effective training system and outreach programs, because in the present form they do not bring the desired results. In this sense, the work has not only a scientific but also an application dimension.

As for comments, there was no literature review on consumer behavior in the context of ensuring food security in the process of purchasing agricultural products. Thus, it cannot be stated unequivocally that the responsibility for various types of threats (e.g. diseases mentioned by the Authors) is the result of improper activity attributable to farmers and / or vendors. It would be worth supplementing this part of study with relevant data. In addition, one can get the impression that the study only points to food safety risks arising from the local sale of food, while this kind of distribution includes many advantages, especially when compared to industrial production.Taking into account the fact that the food sold directly at farmers markets constitutes  the surplus of production made for the agricultural farm's self-consumption, its quality is often higher than in terms of industrial food.

The second remark concerns the methodology. In the "Participants and data collection" part, there was no description of the selection of the sample for the study, methods of conducting surveys and persons responsible for this task. The authors also did not indicate whether the sample size is sufficient to make a conclusions. It can be assumed that due to the limited number of respondents, studies can be treated as a kind of case study. This fact should be emphasized in the article.

From some minor comments, it makes no sense to repeat the same phrases in keywords (like food safety, food safety training etc.).

Author Response

Reviewer 1 comments:

As for comments, there was no literature review on consumer behavior in the context of ensuring food security in the process of purchasing agricultural products. Thus, it cannot be stated unequivocally that the responsibility for various types of threats (e.g. diseases mentioned by the Authors) is the result of improper activity attributable to farmers and / or vendors. It would be worth supplementing this part of study with relevant data. In addition, one can get the impression that the study only points to food safety risks arising from the local sale of food, while this kind of distribution includes many advantages, especially when compared to industrial production. Taking into account the fact that the food sold directly at farmers markets constitutes  the surplus of production made for the agricultural farm's self-consumption, its quality is often higher than in terms of industrial food.

Our response:

 Thank you for your comment. The following information was added to the manuscript.

Enhancing food safety behaviors should not be limited to farmers market managers and vendors alone. Consumer’s behavior also plays a significant role in increasing a risk of foodborne illness. Studies have shown that enhanced consumer food safety practices may lead to a decrease in the risk of foodborne illness at farmers markets [10]. Wolf, Spittler, and Ahem (59) investigated consumers’ perception and the reasons they shop at farmers markets. The authors found that quality and value held the most important attributes for consumers to purchase at farmers markets. Consumers perceive that the produce sold at farmers markets look fresher, are higher-quality, and a better value or reasonably priced, but they do not necessarily consider safety of the products.

The second remark concerns the methodology. In the "Participants and data collection" part, there was no description of the selection of the sample for the study, methods of conducting surveys and persons responsible for this task. The authors also did not indicate whether the sample size is sufficient to make a conclusion. It can be assumed that due to the limited number of respondents, studies can be treated as a kind of case study. This fact should be emphasized in the article.

Our response:

Thank you for your comment. We acknowledge that we lacked clear articulation describing sample selection for the study, methods of conducting surveys, and persons responsible for this task. The following description about the selection of the samples and methods were added to the manuscript.

The survey was disseminated by investigators at farmers market conferences across TX and AR. The sample size was determined based on the margin of error from the confidence interval based on population distributions in the areas where the survey was conducted.   

From some minor comments, it makes no sense to repeat the same phrases in keywords (like food safety, food safety training etc.). 

Our response:

Thanks for feedback, this was corrected in the manuscript and changed to barrier.

Reviewer 2 Report

Thank you for the opportunity to review this article which makes a useful applied contribution to the literature and deserves to be published.  There are really only two things I would recommend:

1.  I want to know more about the types of farmers markets these people work with.  That is, while their demographics are somewhat useful, what is more useful is knowing what their farmers market demographics are.  That is, are workers at more impoverished farmers markets more likely to fail to understand the food safety practices?

2.  There are a few papers in the agricultural economics literature that I think might be interesting to include within the motivation - one of which is related to my above point and one of which makes the case even more strongly as to why this paper has value:

Bellemare, Marc F., and Ngoc Nguyen. "Farmers markets and food-borne illness." American Journal of Agricultural Economics 100.3 (2018): 676-690.

Baur, Patrick, Christy Getz, and Jennifer Sowerwine. "Contradictions, consequences and the human toll of food safety culture." Agriculture and human values 34.3 (2017): 713-728.

Malone, T., and B. Whitacre. "How rural is our local food policy?." The Daily Yonder (2012).

Author Response

Reviewer 2 comments:

Thank you for the opportunity to review this article which makes a useful applied contribution to the literature and deserves to be published.  There are really only two things I would recommend:

 I want to know more about the types of farmers markets these people work with.  That is, while their demographics are somewhat useful, what is more useful is knowing what their farmers market demographics are.  That is, are workers at more impoverished farmers markets more likely to fail to understand the food safety practices?

Our response:

 Thank for the comment. Specific information regarding market demographics were not included; however, the survey respondents were a combination of urban and rural market managers and vendors. Regarding market specifics, the following information was added to the results section.

Therefore, produce was represented as the majority of products sold at surveyed farmers markets. Majority of farmers markets had a U-shaped layout.

There are a few papers in the agricultural economics literature that I think might be interesting to include within the motivation - one of which is related to my above point and one of which makes the case even more strongly as to why this paper has value:

Bellemare, Marc F., and Ngoc Nguyen. "Farmers markets and food-borne illness." American Journal of Agricultural Economics 100.3 (2018): 676-690.

Baur, Patrick, Christy Getz, and Jennifer Sowerwine. "Contradictions, consequences and the human toll of food safety culture." Agriculture and human values 34.3 (2017): 713-728.

Malone, T., and B. Whitacre. "How rural is our local food policy?." The Daily Yonder (2012).

Our response:

Thank you for your suggestions. In order to address the lack of articulation on our part, we added the provided references and the following revisions to our manuscript, as seen below:

A previous study has investigated the relationship between the growing number of farmers markets and the number of cases of foodborne illness in the United States and found a positive correlation between this increase and norovirus-related outbreaks [Ballemare et al., 2018]. This may be due to the lack of handwashing stations and refrigerators accessible at farmers markets.

Ultimately, food safety at farmers markets is achieved by translating behavior into food safety culture [Baue et al., 2017]. This culture should include farmers markets manager, vendor, and consumer. From a socioeconomic perspective, farmer’s markets enhance economic growth in the United States through selling locally grown foods and increasing job opportunities. In addition, since both governmental policies and media support local food, farmers' markets promote economic development for rural and urban communities [Malone & Whitacre, 2012].